# Healthcare Utilization and Adherence to Treatment Recommendations among Children with Type 1 Diabetes in Poland during the COVID-19 Pandemic

**DOI:** 10.3390/ijerph19084798

**Published:** 2022-04-15

**Authors:** Justyna Grudziąż-Sękowska, Kuba Sękowski, Bartosz Kobuszewski

**Affiliations:** 1Centre of Postgraduate Medical Education, School of Public Health, 01-813 Warsaw, Poland; bkobuszewski@cmkp.edu.pl; 2Doctoral School, Law College, Kozmiński University, 03-301 Warsaw, Poland; kuba.sekowski@gmail.com

**Keywords:** diabetes mellitus, type 1, healthcare utilization, pandemic, adherence, DKA

## Abstract

Type 1 diabetes mellitus (T1D) is, next to obesity and asthma, the most common chronic disease in children in Poland. The results of T1D treatment strongly depend on the patient’s compliance with therapeutic recommendations, which entails the use of necessary health services. Based on a retrospective analysis of the data on health services provided in 2016–2020 to over 15.5 thousand patients with T1D in Poland, we assessed the compliance of the actual model of treatment of T1D in children with the current guidelines. It was found that only about 50% of patients received the number of diabetes consultations corresponding to the recognized standards, with about 15% of children with T1D remaining outside the public healthcare system. In the case of many outpatient services (ophthalmological, neurological, mental health), the number of consultations was extremely low—one order of magnitude lower than in general population and dropped even lower in 2020. This shows that the health needs of children with T1D are not being met within the public healthcare system. The COVID-19 pandemic caused significant limitations in access to healthcare in Poland. Compared to the pre-pandemic period there was a significant decrease (−27% compared to 2019) in the number of hospitalizations, and a substantial increase (+22% compared to 2019) in the number of diabetic ketoacidoses (DKA) cases. The proportion of hospitalizations caused by DKA rose to 8.9% compared to 7.3% in 2019.

## 1. Introduction

Type 1 diabetes mellitus (T1D) is an immune-mediated disease. It is a disorder characterized by progressive destruction of pancreatic beta cells, leading to the cessation of insulin secretion and consequent hyperglycemia [1,2]. For more than a century, since the discovery of insulin in 1921, the treatment of T1D has relied on ensuring an external supply of this hormone to maintain a balanced glycemic level [3]. Since the late 20th century, there has been an upward trend in the number of new cases of T1D diagnosed in children and adolescents [4].

The main clinical and social problems associated with T1D are the acute (particularly diabetic ketoacidosis (DKA)) and chronic (nephropathy, neuropathy, retinopathy) complications caused by the disease [5]. Their occurrence generates enormous social and economic costs [6,7]. Preventing the occurrence of complications is one of the main therapeutic challenges in the treatment of T1D.

There is a consensus that modern T1D therapy should include the use of modern medications (rapid-acting insulin), advanced glycemic monitoring technologies (CGM, FGM), and care by various providers (physicians, nurses, educators, etc.) [8]. Current treatment standards also indicate the recommended frequency of follow-up visits and specialist advice (specialist referrals) and the performance of specific tests, including laboratory tests [3,9].

Available studies indicate a significant association of T1D treatment outcome with adherence to established treatment recommendations [10], including regular follow-up visits and laboratory tests [11]. Following these recommendations also positively affects the patient’s psychological well-being and perception of disease burden [12].

The outbreak of the SARS-CoV-2 pandemic resulted in limited access to diagnostics and treatment of many diseases. This phenomenon occurred in many countries and led to inter alia the so-called health debt [13]. In the case of diabetes mellitus, limitations or delays in access to healthcare may increase short- and long-term complications [10,11,12].

Poland is the eighth most populous country in Europe (fifth in the EU). It is also the largest country in Central and Eastern Europe, representative of the former states of the Eastern Bloc. Therefore, epidemiological data from the period before, during, and after the pandemic can show the scale of the challenges in diabetes mellitus management in the region and serve as the basis for planning health policies in this area.

The objectives of this study were (1) to analyze the use of healthcare services by children and adolescents with T1D in Poland, (2) to assess the consistency of the actual frequency of use of specific services with recognized standards and guidelines, and (3) to determine the impact of restrictions resulting from the COVID-19 pandemic on the use of healthcare services by children and adolescents with T1D.

### 1.1. T1D in Poland

In Poland, T1D is, next to obesity and asthma, the most common chronic disease in children [14]. According to the current legal regulations, all children (persons up to 18 years of age) are covered by a universal health insurance scheme in Poland. It means that they have access to healthcare services financed from public funds regardless of the payment of health insurance premiums by their parents/guardians [15,16].

People with diabetes are cared for by diabetologists—doctors who specialize in the diagnosis and treatment of diabetes. In the case of children and adolescents, pediatric diabetologists work in diabetes clinics for the children. In addition to the doctors, diabetes nurses and dieticians also care for the children in these clinics. As part of the medical advice, glycated hemoglobin (HbAc1) levels and other laboratory tests are performed in these clinics [17]. Children with T1D can also benefit from specialist advice (ophthalmologist, neurologist, psychologist, and psychiatrist), which, however, is not part of the activities of the diabetes clinics.

The Polish Diabetes Association (PTD) guidelines for the therapeutic management of diabetes in children and adolescents are consistent with current standards in other developed countries [9]. Unlike in other countries, however, it is recommended that for each new onset of diabetes, the child should be hospitalized in a specialized pediatric diabetes unit, and then should remain under regular, specialized care in pediatric and adolescent diabetes clinics until can be transferred to an adult diabetes clinic. Concerning outpatient care, the PTD formulates requirements to employ diabetes educators and psychologists in addition to physicians, diabetes nurses, and dieticians in the pediatric diabetes clinic. According to the PTD position, visits of a child in the diabetology clinic should take place at intervals of 6–8 weeks, but not less than 4 times a year. The PTD position is not binding for the functioning of the public healthcare system but constitutes only a guideline.

The Polish healthcare system is experiencing many difficulties. Due to the lack of sufficient financial resources and personnel problems (insufficient number of specialist doctors, nurses, and other staff), there are limitations on the availability of certain services (particularly specialist consultations) resulting in extended waiting periods (the so-called “queues/waiting lists” phenomenon) [18].

In the literature, there are numerous examples of studies assessing general health status, metabolic compensation, psychological well-being, and social functioning of children and adolescents with T1D in Poland. However, there is no information on the actual use of healthcare services by children and adolescents.

### 1.2. SARS-CoV-2 Pandemic in Poland—Effect on the Healthcare System

The first case of COVID-19 in Poland was diagnosed on 4 March 2020. The increase in the number of cases, although delayed and slower than in other European countries, resulted in the declaration of an epidemic state by the national authorities and the subsequent introduction of many restrictions and organizational changes in the healthcare system in the second half of March 2020.

During the pandemic period, scheduled admissions were suspended at many hospitals and outpatient (ambulatory care) clinic operations were limited. The restrictions were strictly enforced until June 2020. In the meantime, the possibility of remote medical assistance was slowly implemented. However, telephone consultations were predominantly used, which significantly limited the physician’s ability to accurately diagnose the problem and appropriately guide treatment [19,20].

For patients with T1D, the restrictions made it difficult to have regular follow-up appointments. There was also the reluctance to use healthcare services due to the fear of SARS-CoV-2 infection, which raised serious concerns about possible health consequences [21]. Patients with the first symptoms of T1D were in a particularly difficult situation. The unspecific nature of these symptoms combined with the inability of the physician to examine the patient in person, lead to a delay in the diagnosis of T1D and the development of DKA, and consequently could endanger the patient’s health and life [22,23,24,25].

## 2. Materials and Methods

### 2.1. Study Design

The research utilizes the epidemiological registry-based study approach to determine the number and frequency of various health services used by patients with T1D in Poland. Results were contrasted with established standards and guidelines for T1D treatment in children and adolescents. The data from the pre-pandemic period (years 2016–2019) were compared with the data from the first year of the COVID-19 pandemic.

### 2.2. Source of Data

The research material consisted of data collected by the National Health Fund (NHF). NHF is the state-owned entity responsible for settling the costs of benefits provided under the public health insurance system (universal health scheme). For this purpose, the NHF collects and processes the data on the number and type of services provided to individual insured persons. These data include, among others, information on clinical diagnoses (following the classification contained in ICD-10). However, the NHF does not possess information about the tests performed (if they are not a separate service, but only an element of advice) or their results (e.g., result of HbAc1 determination). All children and adolescents (<19 years old) are entitled to health services financed by NHF.

The structure of the data collected by the NHF enables the determination of the number of individual healthcare services provided in each of the years of the analyzed period, as well as the determination of the number of persons (based on the number of unique PESEL numbers—identifiers assigned to individuals as part of the universal system of electronic population records) to whom they were provided.

For the purposes of population comparisons, the official data on the population of Poland aged under 19 were used. These data are made available to the public by the Central Statistical Office (CSO).

### 2.3. Scope of Data Analysis

The subject of the analysis was information concerning children and adolescents—defined as beneficiaries under 19 years of age (about 6.5–7 million people). T1D patients (ca. 14–16 thousand persons) were distinguished from this group—the identification criterion was a diagnosis E10 according to ICD-10. DKA cases (ca. 860–1,400 persons) were identified based on diagnosis E10.1 according to ICD-10. The search involved the identification of:The number of patients (persons under 19 years of age with diagnosis E.10 according to ICD-10) including (NHF data):
The number of patients at the end of the accounting/reporting period.The number of new patients/diagnoses in the accounting/reporting period.
The number of persons under 19 years of age:
In the general population (CSO data);Beneficiaries of healthcare services provided under the universal health insurance scheme (NHF data).
The number of patients from pt. 1 above who during the accounting/reporting year were provided with (NHF data):
One diabetes consultation;Two diabetes consultations;Three diabetes consultations;Four diabetes consultations;Healthcare services associated with hospitalization in the diabetic ward.
The number of persons (separately from points 1 and 2 above) who during the accounting/reporting year were provided with specific healthcare services (NHF data):
Ophthalmological consultation (examination for retinopathy);Neurological consultation (diagnosis of neuropathy);Mental health advice (psychological/psychiatric care).

Due to the sensitive nature of the collected information, following the regulations on the protection of personal data in force in Poland, the NHF does not provide data allowing aggregation of information on services provided to a specific person (in the form of a database, in which individual records would contain analyzed variables concerning a specific person). For this reason, the analysis of the collected information was performed using descriptive statistics tools.

According to the procedures for verification and control of settlements, data in the NHF system may be corrected by both providers and the NHF. This causes a delay in data availability. It is assumed that after 2 quarters the modification of information in the system may take place only in extraordinary circumstances. In this study, information for the period 2016–2020, as of the end of June 2021, was the subject of analysis.

The anonymous character of the analyzed data resulted in the lack of necessity to obtain consent for participation in the study. Nevertheless, the research program under which the study was conducted received a positive assessment from the IRB of the Centre of Postgraduate Medical Education (no. 501-4-44-28-18).

## 3. Results

### 3.1. Population of Children with T1D

Poland’s population decreased during the period under analysis from 38.437 million at the end of 2015 to 38.265 million at the end of 2020. In contrast, the number of people under 19 years of age showed a slight upward trend. A vast majority of children and adolescents (about 96–97%) have used health services provided under the universal health insurance scheme. The only exception was in 2020 when almost 11% of people under 19 years of age did not use any health benefit under the public system. This represented a several-fold increase from previous years. Children and adolescents with T1D were consistently about 0.2% of the total number of beneficiaries (Table 1).

The number of children and adolescents with T1D who received healthcare services under the universal health insurance system in Poland in particular years showed a steady upward trend throughout the analyzed period (Table 2). It increased from 14,288 persons at the end of 2016 to 15,693 persons at the end of 2020, an increase of more than 9.8%. The highest increase in the number of patients occurred in 2018, at an annualized rate of 3.7%. In the same year, the highest number of new cases of T1D was also diagnosed, with 1486 new cases in nominal terms, corresponding to an incidence of 20.32 cases per 100,000 of the Polish population under the age of 19.

The distribution of the incidence of T1D in the population under 19 years of age benefiting from the public health system is shown in Figure 1.

### 3.2. Out- and Inpatient Care for Children with T1D

During the study period, children with T1D most frequently received diabetes consultations provided on an outpatient basis (Table 3). These services (diabetes visits) were used by more than 85% of the population with T1D under the age of 19 each year. This value remained relatively stable throughout the analysis period, with a slight increase in 2020 when outpatient services (at least one diabetes consultation) were used by more than 88% of children with T1D. The average (weighted average) number of consultations was, respectively: 2.84 in 2016, 2.84 in 2017, 2.83 in 2018, 2.79 in 2019, and 2.87 in 2020. The change in the distribution of children receiving one, two, three, or four or more diabetes consultations, respectively, is shown in Figure 2.

The number of services provided in the inpatient setting (hospitalizations) in 2016–2019 oscillated between 6.9–7.5 thousand cases (corresponding to about 47–50% of the population of children with T1D in a given year) and decreased to about 5.5 thousand cases (35.4% of the population of children with T1D) in 2020. (Table 4). Most hospitalizations occurred in pediatric diabetes wards (approximately 17% to approximately 23% of total hospitalizations), general pediatric wards (approximately 13% to approximately 17% of total hospitalizations), and pediatric endocrinology wards (approximately 7% to approximately 15% of total hospitalizations). The remaining hospitalizations occurred in other wards (Figure 3).

### 3.3. DKA

The collected data show a large time variability in the number of DKA cases diagnosed in children under 19 years of age by a quarter from 2017 to 2020 (Figure 4). During the period Q1 2017. In Q1 2020 there was a steady slow increase in the number of DKA cases in children. This was interrupted in Q2 2020 when there was a decline in the number of cases. followed by a spike in DKA in Q3 2020. The sum of the quarterly number of DKA hospitalizations is higher than the number of patients with this diagnosis each year. This indicates multiple DKA incidences in some patients.

### 3.4. Utilization of Other Health Services

Data collected by the NHF indicate an extremely low number of outpatient ophthalmic, neurological, and mental health services provided to patients with T1D. In patients with T1D, such consultations occur ten times less frequently than in the general population. The data in Table 5 do not include consultations given to these patients during hospitalization (as part of inpatient care).

## 4. Discussion

Epidemiology of T1D in Poland is characterized by comparable incidence and slightly lower prevalence as in other European countries [26]. The changes IN these rates over time are following the prognoses [27]. The current standards of management—in the form of PTD recommendations—are convergent with those used in developed countries. The distinguishing element of PTD recommendations is the promotion of an inpatient model of care (hospitalization) in the situation of T1D diagnosis in a child [9]. In addition, periodic/cyclical hospitalizations have been used in therapeutic practice for many years aimed at diagnostics, “equalization” of the disease, and education of the patient and his/her family [28].

### 4.1. Outpatient Care Utilization

In accordance with the first objective of this study, we found that the vast majority (approximately 85%) of patients with T1D annually receive diabetes counselling provided in outpatient care units under the universal health insurance scheme. The number of outpatient consultations is one of the primary components of assessing patient adherence to treatment recommendations [29], as well as a frequently used predictor of treatment outcomes [30,31].

In the case of over a half of children and adolescents with T1D in Poland (51.7%), the frequency of these visits corresponds to or even exceeds the minimum PTD recommendations in this respect (at least 4 visits per year). This indicates a high level of patients’ (their guardians) involvement in the therapeutic process [32]. This is despite the fact that the guardians of children and adolescents with T1D declare difficulties in arranging an appropriate number of diabetes consultations [33]. One factor affecting the regularity of follow-up visits may be of a non-cyclical nature. The orders (prescriptions) for supplies for insulin pump accessories (infusion sets. insulin cartridges, etc.) and CGM systems (sensors/electrodes, transmitters, etc.) are issued for 3 to 12 months and require renewal at the subsequent visit.

Of concern is the suboptimal use of diabetes care by the rest of the population of children and adolescents with T1D, including a relatively high number of patients receiving two or fewer consultations per year (32.5%). This value is higher than that obtained in other studies [34]. The available findings suggest that inadequate frequency of treatment follow-up may have a negative impact on treatment outcomes [11,29,30]. This is particularly true for approximately 11–15% of the population who do not receive any diabetes counselling.

For the remaining outpatient services that should be provided to children and adolescents with T1D according to current treatment standards, the frequency of use of those examinations and consultations that do not fall under diabetes counselling was assessed. These included ophthalmological advice (including funduscopic examination), neurological advice, and mental health services (psychological and psychiatric advice). These services are important in the prevention of T1D complications [35,36].

The comparison of the obtained results with the current recommendations of PTD (e.g., follow-up ophthalmological examination within 5 years from the diagnosis of T1D) and the available results of analyses of the functioning of the care of children and adolescents with T1D in other countries [37], indicates a very small, even symbolic, degree of fulfilment of these health needs.

There are many possible reasons for such a low degree of adherence by patients (their caregivers) to recommendations for regular eye and neurological examinations. This may result from objective reasons—lack of knowledge about the recommended frequency of examinations and consultations, or subjective ones—underestimation or neglect of the problem by the patient/carer [38,39]. In the realities of the Polish public healthcare system, the limited availability of particular services may also be relevant—which refers especially to outpatient specialist care. This corresponds to the results of studies assessing the problems of families of children and adolescents with T1D in Poland [33]. Nevertheless, the phenomenon of low use of outpatient services is worrying and requires further analysis. They should address both the health effects and causes of suboptimal therapy in children and adolescents with T1D.

### 4.2. Inpatient Care Utilization

One of the primary causes of hospitalization of children with T1D is the occurrence of DKA. According to an extensive analysis by Usher-Smith et al., the incidence of DKA varies widely across countries (from 12.8% to 80%) and is inversely associated with gross domestic product, latitude, and background incidence of T1D [40]. In children with established T1D, DKA episodes occur at a rate of 1% to 8% per year [41].

The number of hospitalizations observed in the study was much higher (on average about 6900 cases per year) and corresponded to about 47–50% of the population of children with T1D in a given year. It was therefore much higher throughout the study period than observed in other studies [40,41,42,43]. This is only partly explained by the practice—recommended by PTD—of hospitalizing all children diagnosed with T1D (on average about 1.320 new cases per year), or the necessity to treat DKA cases. This points to the other above-mentioned causes of hospitalization, including the practice of “educational” hospitalizations indicated in the literature [28]. Such planned hospitalizations provide an opportunity to perform necessary diagnostic tests and consultations, while the availability of such consultation in the outpatient setting is limited [18,33]. This finding is consistent with the results indicating a small number of ophthalmological, neurological, and mental health consultations in the outpatient care, which were used by children with T1D during the study period.

### 4.3. COVID-19 Pandemic’s Influence on Adherence and Healthcare Utilization

The development of the COVID-19 pandemic and the changes in healthcare systems in response to it have raised a number of concerns about the impact of these measures on the health of children with T1D. There have been pointed out possible risks from limited access to healthcare [44], impaired glycemic control during lockdown [45,46], and increased risk of DKA onset even for people with T1D who were not infected with SARS-CoV-2, most likely due to delays in accessing care [47,48]. Recently published studies however did not demonstrate the deterioration of T1D patients’ health that could be attributed to the COVID-19 lockdowns [49,50,51,52]. Some studies have observed improved glycemic compensation in adults [53] and children [54] with T1D while they were at home due to lockdowns. However, these studies were not population-based, and their results cannot be generalized without reservation to the entire population of children with T1D.

In the present study, it was possible to observe significant changes in the number and structure of health services provided to children with T1D in the period preceding the outbreak of the COVID-19 pandemic and during the restrictions and organizational changes in the functioning of the healthcare system. The introduction of lockdown in Poland (second half of March–June 2020) corresponds with a break in the previously sustained upward trend in the number of DKA cases. After a periodic decline in the number of DKA cases in Q2 2020, there was a sharp increase in their number in Q3 2020, and the total number of DKA in 2020 was almost 22% higher than in the previous year. This may indicate a phenomenon of deferred demand for health services due to fear of SARS-CoV2 infection or because of difficulties in obtaining them, as observed in other health systems [55]. At the same time, a study conducted at the Warsaw Medical University Hospital during the restriction period found a 12 percentage point increase in the incidence of DKA in the course of newly diagnosed type 1 diabetes in children along with an increase in the proportion of severe cases [21]. An increase in severe DKA (from 36.1% in 2019 to 44.3% in 2020) was also observed in a study of Italian children by Rabbone et al. [48]. However, it should be noted that Poland experienced a significant (by approximately 27% relative to 2019) decrease in hospitalizations of children with T1D throughout 2020, resulting in an increase in the proportion of hospitalizations due to DKA.

The number of hospitalizations due to new T1D diagnoses remained relatively unchanged, the reduction in the total number of hospitalizations resulted from the significantly lower number of hospitalizations due to other causes, mainly planned treatment—hospitalizations for diagnostics and consultations. In the long term, this may cause delays in the diagnosis of T1D complications, as access to such diagnostics and consultations in outpatient care is limited.

In terms of the use of outpatient services, it was observed that the average annual number of diabetes consultations has not changed despite the pandemic. The average number (ca. 2.8 consultations/patient/year) was similar to the values observed in the pre-pandemic period. At the same time, the differences between the number of visits used by different groups of patients have widened—the group of patients taking four or more visits per year has increased (to 51.7% of the population of children with T1D), as well as those making the fewest visits (1 or less)—20.8%. In 2020, remote medical visits (telemedicine) were implemented on an unprecedented scale in Poland. Available studies indicate the high usefulness and popularity of this form of service to patients with T1D [56]. However, their introduction may disadvantage underprivileged groups—people who are less familiar with or lack access to modern technologies [57].

### 4.4. Practical Implications

Although most of the young T1D patients undergo the recommended number of diabetes consultations per year, there is a significant group with very little or no contact with health professionals. The significant percentage of children who do not receive any diabetes counselling or undergo only one consultation per year must be of concern. Pandemic-related limitations have made this problem even more disquieting. Health care practitioners should be aware of this phenomenon and should try to identify reasons for the nonadherence of some patients.

The data collected indicate negligible use of ophthalmological, neurological, and mental health consultations by children with T1D. Some form of compensation is carried out in the form of hospitalizations, as half the population of children with T1D in Poland undergo them each year. However, this is a costly solution, and prone to disturbances, as proved by the current pandemic.

Therefore, special attention should be given to the outpatient ophthalmological consultations, as they should be a part of regular check-ups in T1D, while the number of patients that actually use such consultations is marginal.

### 4.5. Study Limitations

This study was based on data gathered and processed in the NHF’s computer system for primarily financial reasons. Analyzed data were provided (entered into the system) by various health service providers acting under the universal health insurance scheme, therefore it did not reflect health services provided outside the public health system.

## 5. Conclusions

The results of this registry-based study indicate inadequate utilization of health services by the population of T1D children and adolescents in Poland and a substantial decrease in the number of such services used during the COVID-19 pandemic. The relevance of the study’s results is enhanced by the fact that analyzed data (kept in the NHF’s registry) show the use of healthcare services by over 96% of children and adolescents in Poland (approx. 89% during the pandemic).

The degree of adherence to the recommendations regarding the type and frequency of use of health services is relatively high only in the case of the use of periodic diabetes consultations. Half of the patients have such consultations regularly, in the number corresponding to the current PTD recommendations. At the same time, the number of ophthalmological, neurological, and mental health consultations is extremely low and does not correspond with current standards of T1D treatment.

The impact of the COVID-19 pandemic on the health status and use of healthcare services by children with T1D is difficult to assess precisely at this stage. However, the observed health deterioration of newly diagnosed T1D cases manifested in higher DKA numbers may be accompanied by the increased prevalence of long-term complications due to cancelled or postponed consultations. This risk should be included in the diabetes mellitus management strategies, health policies, and contingency plans at the national level.

## Figures and Tables

**Figure 1 ijerph-19-04798-f001:**
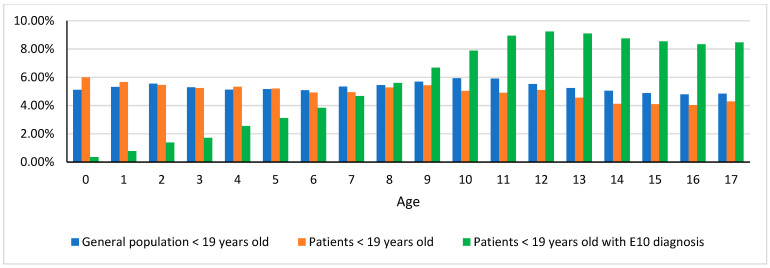
Age distribution of the population of patients with T1D compared to the general population of persons under 19 years of age and the population of persons under 19 years of age receiving services from the public healthcare system.

**Figure 2 ijerph-19-04798-f002:**
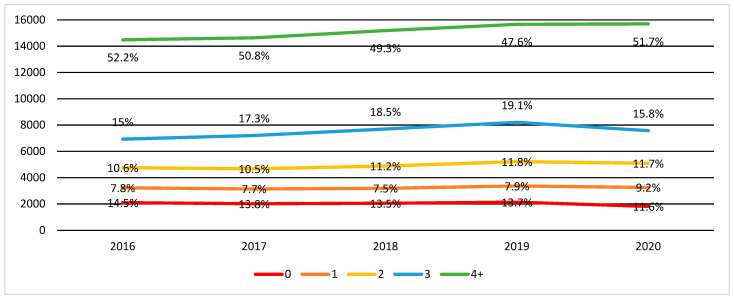
The number of diabetes consultations received by children with T1D by year—presented as absolute number (left scale) and percentage from the population of children with T1D in a given year.

**Figure 3 ijerph-19-04798-f003:**
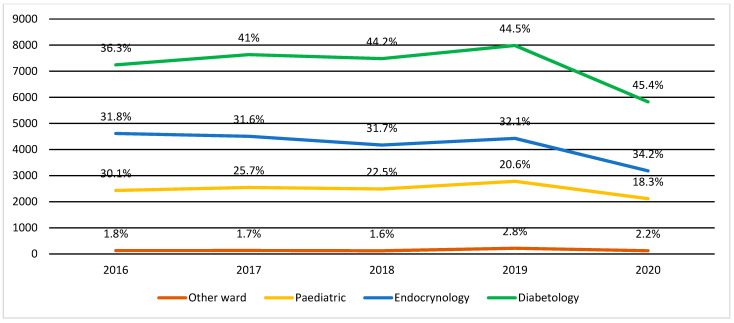
Distribution of hospitalizations of children with T1D by type of hospital unit by year—presented as absolute number (left scale) and percentage from the total number of hospitalizations of children with T1D in a given year.

**Figure 4 ijerph-19-04798-f004:**
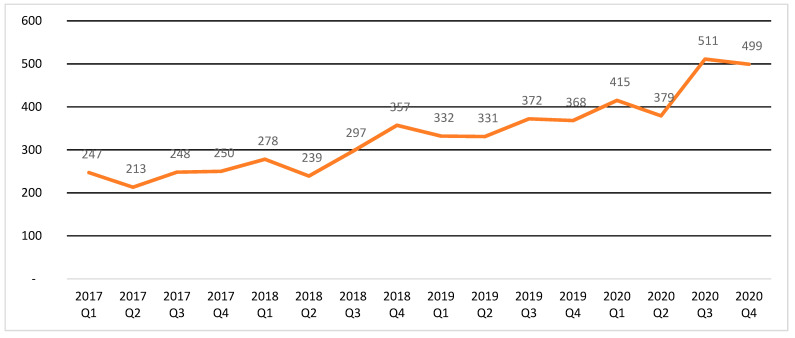
Number of DKA cases by a quarter from 2017 to 2020.

**Table 1 ijerph-19-04798-t001:** The population of people under 19 years of age in Poland, including those using services of the public healthcare system.

Year	The General Population (Pop.) < 19 Years Old	Public Health System Patients < 19 Years Old		Public Health System Patients with T1D (%)
		No.	% of Pop.	% of All Patients < 19 Years Old
2016	7,286,480	7,070,670	97.0%	0.2%
2017	7,299,996	7,088,663	97.1%	0.2%
2018	7,311,538	7,085,435	96.9%	0.2%
2019	7,314,617	7,056,062	96.5%	0.2%
2020	7,307,098	6,513,836	89.1%	0.2%

**Table 2 ijerph-19-04798-t002:** Prevalence of T1D in Poland in persons under 19 years of age—beneficiaries of the public healthcare system.

Year	Total Number of T1D Cases at the End of Year ^1^	Newly Diagnosed Cases— Nominal (Yearly) ^2^	Newly Diagnosed Cases—Per 100 Thousand Population < 19 Years Old (Yearly) ^2^
2016	14,288	1228	16.85
2017	14,634	1407	19.27
2018	15,182	1486	20.32
2019	15,663	1346	18.40
2020	15,693	1135	15.53

^1^ Number of persons aged < 19 years for whom any publicly funded healthcare service with E10 diagnosis according to ICD-10 was settled. ^2^ Number of people aged < 19 years old for whom any publicly funded healthcare service with E10 diagnosis according to ICD-10 was settled for the first time.

**Table 3 ijerph-19-04798-t003:** Frequency of use of diabetes outpatient services by individuals under age 19 with T1D by year.

Health Service Used/Yearly	2016	2017	2018	2019	2020
	Number of Patients Diagnosed with E10 (Absolute Number and Percentage from the Population of T1D Patients < 19 Years of Age at that Time)				
1 diabetes consultation	1107	1126	1137	1229	1441
	(7.7)	(7.7)	(7.5)	(7.8)	(9.2)
2 diabetes consultations	1355	1531	1696	1849	1834
	(9.5)	(10.5)	(11.2)	(11.8)	(11.7)
3 diabetes consultations	2170	2531	2810	2989	2480
	(15.2)	(17.3)	(18.5)	(19.1)	(15.8)
4 or more diabetes consultations	7553	7429	7483	7455	8120
	(52.9)	(50.8)	(49.3)	(47.6)	(51.7)
No consultations	2103	2017	2056	2141	1818
	(14.7)	(13.8)	(13.5)	(13.7)	(11.6)

**Table 4 ijerph-19-04798-t004:** Utilization of inpatient diabetes services by individuals under age 19 with T1D by year.

Health Service Used/Yearly	2016	2017	2018	2019	2020
	Number of Patients Diagnosed with E10 (Absolute Number and Percentage from the Population of T1D Patients < 19 Years of Age at that Time)				
Hospitalization for:	6924	7270	7131	7595	5548
	(48.5)	(49.7)	(47.0)	(48.5)	(35.4)
Diabetes care unit for children	2629	3127	3308	3556	2644
	(18.4)	(21.4)	(21.8)	(22.7)	(16.8)
Pediatric endocrinology ward	2181	1960	1683	1645	1065
	(15.3)	(13.4)	(11.1)	(10.5)	(6.8)
Pediatric ward	2305	2415	2369	2564	1993
	(16.1)	(16.5)	(15.6)	(16.4)	(12.7)
Other wards	130	133	122	220	125
	(0.9)	(0.9)	(0.8)	(1.4)	(0.8)
Number of patients diagnosed with ketoacidosis ^1^	n/a	860	1038	1151	1404
		(5.9)	(6.8)	(7.3)	(8.9)
(Diagnosis E10.1 according to ICD-10) ^2^	n/a	n/a	n/a	981	1153
				(6.3)	(7.3)

^1^ Diagnosis data collected using a 4-digit code according to ICD-10 collected from 2017. ^2^ Accompanying diagnosis data collected using a 4-digit code according to ICD-10 collected starting from 2019.

**Table 5 ijerph-19-04798-t005:** Number of individuals receiving individual outpatient services (ophthalmological, neurological, mental health consultations) separately for T1D patients and the general population of individuals under 19 years of age.

Health Service Used/Yearly	2016	2017	2018	2019	2020
	Total Number of Patients/Number of Patients with Diagnosis E10 (Absolute Number and Percentage from the General Population and the Population of T1D Patients < 19 Years of Age)				
Ophthalmological consultation with fundus examination	4027/2 (0.055/0.014)	4379/0 (0.060/0)	4037/2 (0.055/0.015)	4278/3 (0.058/0.019)	3272/0 (0.045/0)
Neurological consultations	176,319/35 (2.42/0.2450)	171,533/42 (2.35/0.287)	163,568/40 (2.237/0.263)	156,102/45 (2.134/0.287)	130,122/33 (1.781/0.210)
Mental healthcare services:					
Medical diagnostic consultation	50,750/3 (0.696/0.021)	49,966/3 (0.684/0.021)	49,393/9 (0.676/0.059)	51,339/8 (0.702/0.051)	40,224/9 (0.550/0.057)
Medical therapeutic consultation	51,042/5 (0.701/0.035)	52,358/7 (0.717/0.048)	53,292/7 (0.729/0.046)	52,093/8 (0.712/0.51)	51,419/14 (0.704/0.089)
Psychological diagnostic consultation	26,382/2 (0.362/0.014)	24,600/1 (0.337/0.007)	23,068/4 (0.316/0.026)	24,797/2 (0.339/0.013)	18,196/3 (0.249/0.019)
Psychological therapeutic consultation	28,107/0 (0.386/0)	27,876/1 (0.382/0.007)	27,215/2 (0.372/0.013)	28,191/1 (0.385/0.006)	27,840/2 (0.381/0.013)

## Data Availability

The data presented in this study are available on request from the corresponding author. The data are not publicly available due to technical reasons.

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
