# Peer review of "Healthcare Utilization and Adherence to Treatment Recommendations among Children with Type 1 Diabetes in Poland during the COVID-19 Pandemic"

_ijerph, 2022, doi:10.3390/ijerph19084798_

Round 1

Reviewer 1 Report

I think that the work is well designed and presented and is of interest to IJERPH readers.

However, I believe that the data that appear in table 5 deserve a more detailed explanation. If I interpret it correctly, the ophthalmological consultation with fundus examination, neurological consultations and consultations with mental health care services are, for example, in 2020, 10 to 30 times less frequent in T1D patients than in the general population of individuals under 19 years of age. Although the authors comment on these surprising results, I think they deserve some special comment.

I also think that the work would improve if the results could be presented, at least some of them, separated by male and female

Author Response

Dear reviewer,
thank you for your interest in our paper and the comments made.
As regards out-patient care utilization by T1D population in Poland results we obtained are shocking, but we also think that they are a true novelty, and provide a starting point for further research. Current literature does not offer any viable explanation to that fact, therefore, we do not want to speculate on this.
We have, however, rephrased the paragraph referring to those findings (lines 307-317).
As regards the more in-depth presentation of data (e.g. separation of results on a gender basis) unfortunately this could not be done. Data provided by the National Health Fund were aggregated and anonymized, thus no demographic distinctions were possible.

Best regards
Justyna Grudziąż-Sękowska

Reviewer 2 Report

In this manuscript, the author has analyzed the use of health care services by children and adolescents with T1D in Poland. Based on the National Health Fund data, the author has compared the number of T1D cases from 2016 to 2020 and the yearly health service time for patients under 19. In the meantime, the author has studied the DKA cases from 2016 to 2020, trying to reveal a relationship between the decreased T1D hospitalization rate and DKA cases. 

The study was based on a reliable data source. Generally, this manuscript looks good. The author, however, also needs to provide a more detailed analysis. For example, for the patient who terminated the consultation in 2020 or did not get enough consultation, I am curious about their consultation number in 2019. I am not sure whether this kind of information can be tracked down, but it tells the use of health care services in pandemics. 

Author Response

Dear reviewer,
thank you for your interest in our paper and the comments made.
As regards the more in-depth presentation of data (e.g. tracking individuals or groups of patients) the scope of analysis was vastly limited by the type and form of data provided by the National Health Fund.
Those data were aggregated and anonymized, thus no demographic distinctions or tracing of individuals/groups (subcohorts) were possible. All year-to-year comparisons had to base on general numbers for given years.
Therefore, it is hard to judge whether it was the same group of patients who received a suboptimal number of follow-up diabetic consultations during the analyzed period. 
Similarly, available data did not allow the assessment of the actual number of such consultations were received by given individuals in that period.
We hope that our work (especially findings of actual low utilization of the out-patient care) will provide a starting point for further research on that topic. 

Best regards
Justyna Grudziąż-Sękowska

Reviewer 3 Report

This study aimed to evaluate the health care utilisation and adherence to treatment recommendations among children with type 1 diabetes in Poland during  COVID-19 pandemic.

the topic could also be interesting but the paper needs to be completely revised before publication.

1. the readability of the paper is very poor. Many paragraphs are unclear. Also the abstract does not make clear what the objectives of the paper are and is very confusing.

2. The origin of the data for the covid period is not very clear. It seems objectively that the authors have artificially merged data from different sources between before and after covid with no apparent homogeneity.

3. The summary of the paper is very obvious. The covid restrictions reduced the frequency of children with type 1 diabetes mellitus in centres for the treatment of the disease. We could not have expected anything different. The authors should enrich the paper with some consideration of, for example, the effects of this restriction on glycaemic control in these patients by means of glycosylated haemoglobin values.

Nutrition and the different lifestyle had a great influence on the habits of these patients and also on the metabolic effects.  A paragraph on this, perhaps with updated data, might be useful. 

Author Response

Dear reviewer,
thank you for your interest in our paper and the comments made.
Ad. 1.
To improve readability, many paragraphs were rephrased many paragraphs and too-long sentences were shortened.
 Ad. 2.
All the data analyzed were obtained from the National Health Fund - a sole public institution in charge of financing health services provided under the public health insurance system. Therefore, such data can be treated as complete (representing the vast majority of health services provided in Poland, as over 95% of the population is entitled to such services). Unfortunately, NHF's data do not allow for more in-depth comparisons, that require access to the health history of individual patients (e.g. to analyze the demographics of patients who received the suboptimal number of out-patient services).
Information on the source of data was presented in a separate paragraph (lines 123-139).
Ad. 3.
The findings regarding the influence of the COVID pandemic on health care utilization are indeed in-line with the results presented in other papers.
We believe, however, that there is a novelty in our work, as it was based on data gathered on the whole health system level (opposed to researchers conducted in certain hospitals/outpatient clinics). Moreover, our paper relates to the situation in a Central/East European country, while other publications originate from West Europe or other countries.
We agree that a more in-depth analysis of pandemic consequences in Poland is required (including the effects on glycaemic control). This, however, was beyond the scope of the study.
We hope that our work (especially findings of actual low utilization of the out-patient care by patients with T1D) will provide a starting point for further research on that topic. 

Best regards
Justyna Grudziąż-Sękowska

Round 2

Reviewer 2 Report

Although the author did not make many changes in the manuscript and insisted the data source limited the deeper thinking, I think this manuscript can still offer some information about the T1D patient during the pandemic. As the author mentioned, it is a good starting point. 

Author Response

Dear reviewer,

thank you for your comments.
We made some additional changes in line with your sugestions in the Introduction, Study Design and Summary.

Best regards
Justyna Grudziąż-Sękowska

Reviewer 3 Report

The authors have made some slight modifications. The shortcomings in the rationale and structure of the paper remain. The data presented are very obvious: with the pandemic, the number of TDM1 patients attending hospital centres has reduced. Probably with the end of the pandemic everything will return to normal. There was no need for this paper to know this news, which can also be found in the newspapers. 

1. the readability of the paper is very poor. Many paragraphs are unclear. Also the abstract does not make clear what the objectives of the paper are and is very confusing.

2. The origin of the data for the covid period is not very clear. It seems objectively that the authors have artificially merged data from different sources between before and after covid with no apparent homogeneity.

3. The summary of the paper is very obvious. The covid restrictions reduced the frequency of children with type 1 diabetes mellitus in centres for the treatment of the disease. We could not have expected anything different. The authors should enrich the paper with some consideration of, for example, the effects of this restriction on glycaemic control in these patients by means of glycosylated haemoglobin values.

Nutrition and the different lifestyle had a great influence on the habits of these patients and also on the metabolic effects.  A paragraph on this, perhaps with updated data, might be useful. 

Author Response

Reply was provided in a separate Word file.
